# Study on Stratified Settlement and Weak Reflectivity Fiber Grating Monitoring of Shield Tunnel Crossing Composite Strata

**Fucai Zhao** [1,2], **Xingli Lu** [2,*], **Hongbing Shi** [3], **Bin Liu** [2], **Shaoran Liu** [3], **Kaohong Dai** [4] **and Ying Fan** [1]

1   School of Civil Engineering Architecture and the Environment, Hubei University of Technology, Wuhan 430068, China
2   Wuhan Institute of Rock and Soil Mechanics, Chinese Academy of Sciences, Wuhan 430064, China
3   China Construction Infrastructure Co., Ltd., Shenzhen 518000, China
4   China Construction Communications Engineering Group Co., Ltd., Shenzhen 518000, China
*   Correspondence: xllu@whrsm.ac.cn

**Abstract:** This paper proposes a set of field test technology system for layered settlement of composite strata based on weak reflectivity fiber Bragg grating sensing technology based on the shield project of "Keyuan Station ~ Shenzhen University Station" section of Shenzhen Metro Line 13, and through the comparison and verification of three-dimensional numerical simulation and field monitoring, the law and distribution characteristics of disturbance settlement of ground surface and overlying strata during shield tunneling are systematically analyzed, and the vertical and horizontal zoning (layer) system for the spatial and temporal evolution of layered settlement of composite strata during shield tunneling is constructed. On this basis, the targeted settlement control technical measures and recommendations are proposed. The findings show that the weak reflectivity fiber grating sensing technology can better perceive the evolution law and distribution characteristics of vertical and horizontal settlement of composite strata caused by shield tunneling, which is in good agreement with the numerical simulation results, and has the advantages of automation and high precision, it can be used as a supplement and alternative method for traditional measurement methods. The stratum deformation is small and layered settlement is not obvious in shield approaching stage (−5D~0), after shield crossing and shield tail falling (0~3D), the stratum is the longitudinal main deformation zone of shield tunneling disturbance, and the influence range of the whole tunneling disturbance is about (−1D~3D). Meanwhile, according to the influence degree of shield tunneling disturbance, the overlying strata of the tunnel can be divided into main disturbance layer and secondary disturbance layer, and the main disturbance layer is located in the range of 0.5D above the tunnel. In addition, based on the different stages of shield tunneling and the vertical and horizontal zoning (layers) of existing structures such as buildings (structures), the settlement control measures and suggestions are proposed. The research results demonstrate the feasibility of weak reflectivity fiber grating for distributed and continuous strata monitoring. It has important guiding value for improving the understanding of settlement law produced from shield construction in composite strata and analyzing and predicting potential risks resulting from shield construction. It also provides reference value for future subway design and construction.

**Keywords:** shield tunnel; composite strata; layered settlement; weak reflectivity fiber grating; numerical simulation; field monitoring

## 1. Introduction

With the advent of modern urban subway engineering, the shield tunnel construction technique has also entered a period of rapid development. At the same time, an increasing number of issues in shield tunnel construction have emerged, the most essential of which is the safety problems caused by shield construction to stratum disturbance. Research showed

that shield construction disturbance not only affects the surface road and surrounding buildings (structures) but also affects and even damages various structures below the ground, as well as the internal lining structure of the tunnel deformation and others [1–5]. Many construction processes with different degrees of surface subsidence, building inclination, underground pipeline damage, and others have occurred, severely compromising the safety of subway construction and inflicting enormous economic losses [6–10].

In recent years, using theoretical analysis [11–13], model tests [14–18], field tests, and other methods [19–21], scholars have conducted extensive research on ground deformation caused by shield disturbance and have obtained some valuable results [22]. For a single soft soil layer, the prediction of surface subsidence during tunnel excavation was carried out [23]. A prediction was made on the deformation amplitude of stratum disturbance during the excavation of a shallow tunnel after considering various influencing factors, based on this research, a theoretical test on the settlement law caused by the tunnel was performed, and the tunnel settlement gap parameters were calculated based on the settlement caused by tunneling in a specific case [24]. The model test utilizes the downscaling structure and centrifuge facilities, the deformation and stress characteristics of the tunnel under various overburden depths were analyzed by applying the acceleration required by the research institute [25–28]. Field tests can be achieved on a real-time collection of sensor data to reflect the physical-mechanical behavior of the tunnel's composite strata during shield construction [29–33]. Through the analysis of the monitoring data, the potential hazards caused by the settlement of the overlying stratum of the tunnel can be evaluated. The monitoring data can also be employed to predict the stratum deformation of the pipe section at various tunnel positions. With the growth of optical fiber sensing technology, weak optical fiber sensing technology has accelerated. Compared to traditional monitoring methods such as single-point or multi-point displacement meters, It has the advantages of relatively small disturbance, temperature compensation function and remote monitoring [34–36]. The advantages of small size, lightweight, high sensitivity, and high precision have been widely used in practical projects, including mines, bridges, foundation pit and other practical engineering, and can achieve quasi-distributed and distributed measurements [37–39]. Large-scale engineering problems can be calculated with great efficiency and high accuracy using numerical simulation. In conjunction with a more field-appropriate monitoring method is proposed to improve the accuracy and reliability of analytical results [40–48].

In this paper, the shield project of "Keyuan Station~Shenzhen University Station" section of Shenzhen Metro Line 13 is taken as the engineering background, based on the weak reflectivity fiber grating sensing technology, the field test of layered settlement of composite stratum is carried out. Through the comparison and verification of three-dimensional numerical simulation and weak reflectivity fiber grating field monitoring, the law and distribution characteristics of disturbance settlement of ground surface and overlying strata during shield tunneling are systematically analyzed, and the vertical and horizontal zoning (layer) system of layered settlement of composite stratum with time and space evolution of shield tunneling is constructed. On this basis, the technical measures and suggestions for layered settlement control of composite strata are put forward. The research results have essential guiding significance for improving the cognition of settlement law caused by shield construction in composite strata and the risk analysis and control of existing buildings (structures) in shield construction. This study provides a valuable reference for engineering construction under similar circumstances.

## 2. Engineering Background

### 2.1. Project Overview

This paper relies on the shield project of "Keyuan Station ~ Shenzhen University Station" section of Shenzhen Metro Line 13. The subway tunnel shield project (mileage Z/YDK4 + 562.489-Z/YDK4 + 805.489) is two single-line tunnels repaired separately on the left and right lines with a total length of 243 m, and an average distance between the

left and right lines of the tunnel of 11 m, as shown in Figure 1. The shield segment is a single-layer assembled lining with an outer diameter of 6.0 m and an inner diameter of 5.0 m. There are 6 segments exist in each ring with a thickness of 0.5 m each. In the early stage, the right line has been fully penetrated, and the settlement has stabilized. Therefore, this paper only investigates the stratum settlement law during the penetration of the left line shield tunnel and assumes that the right line does not affect the results, see Figure 1a.

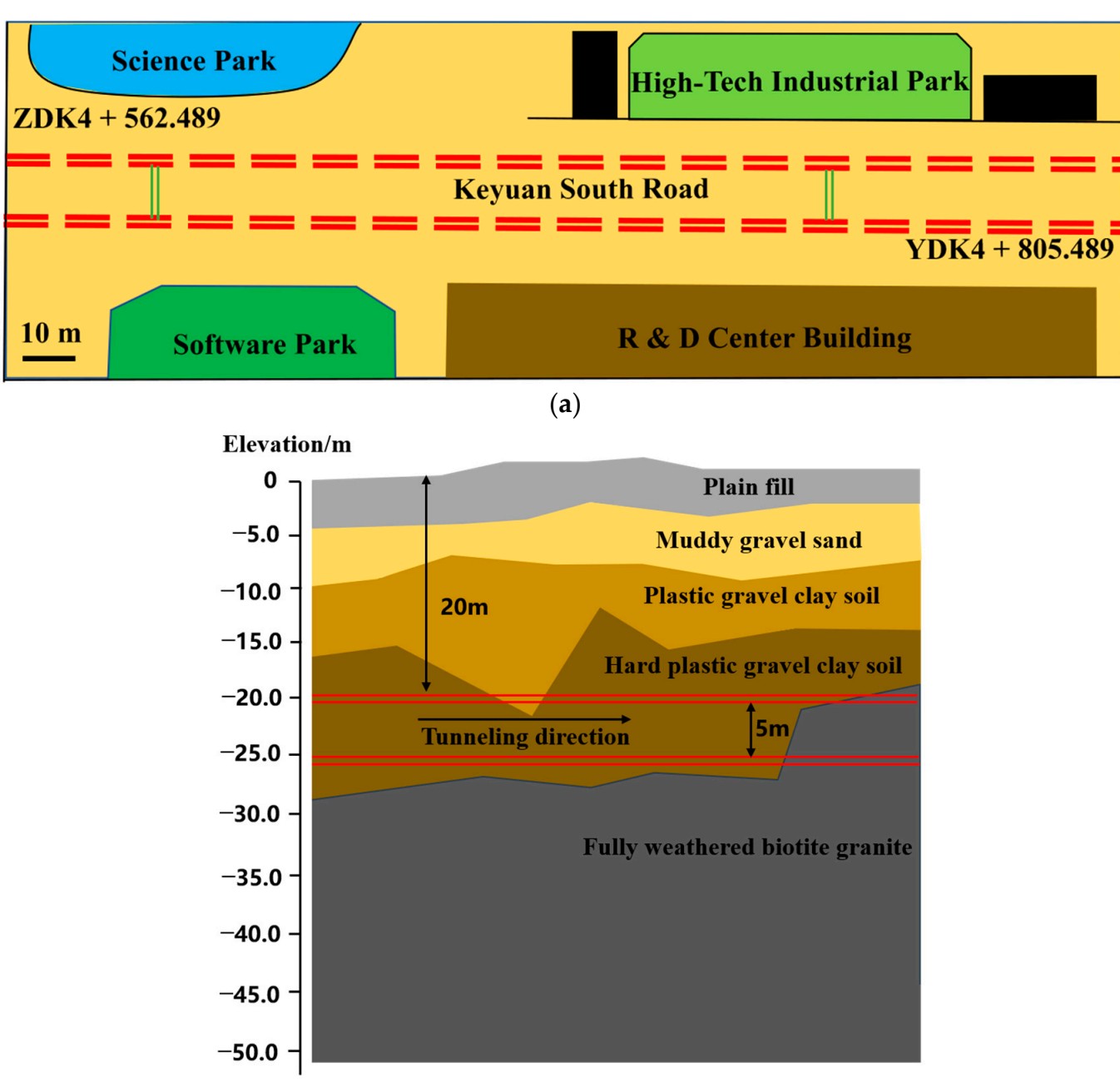

**Figure 1.** Engineering environment diagram. (**a**) A schematic view of the study site. (**b**) Stratum structure diagram.

The shield section passes through a complex and changeable stratum. The stratum from top to bottom consists of a plain fill layer, a muddy gravel sand layer, a plastic gravel clay layer, a hard plastic gravel clay layer, and fully weathered biotite granite. The tunnel body is mainly located in hard plastic gravel clay soil. The tunnel mainly passes

through plastic gravel clay soil, hard plastic gravel clay soil, and fully weathered biotite granite. The average overburden thickness of the shield is 20 m, see Figure 1b. The strength characteristics, compressibility, permeability, and sensitivity of interval soft soil muddy affect the safety of engineering construction and long-term operation and cause great difficulties and risks in the construction and operation of rail transit.

### 2.2. Field Monitoring Layout

In view of the complex and changeable characteristics of the composite strata in the "Keyuan Station~Shenzhen University Station" section, this paper uses a quasi-distributed fixed-point grating monitoring system based on weak reflectivity Bragg fiber grating (FBG) to monitor the layered settlement of the strata. The quasi-distributed fixed-point grating monitoring system adopts a distributed fixed-point grating fiber optic cable, which is an internal fixed-point design, and can realize spatial discontinuous non-uniform strain segmentation measurement. It has excellent coupling for stratum deformation monitoring and can realize compression and tensile deformation measurement.

The layout of the field monitoring system is shown in Figure 2. Seven surface monitoring points and three stratum layered settlement monitoring holes are set respectively. The interval between adjacent surface monitoring points is 3.0 m, and the stratum layered settlement monitoring holes are set as follows: hole 1 # and 3 # are 22 m deep, hole 2 # is 20 m deep. The research shows that the monitoring data near the surface is greatly affected by temperature, and the temperature compensation method can effectively reduce the influence of temperature on the deformation of shallow soil. In this field test, a double-ended measurement and testing system composed of 1.0 m fixed-point grating optical cable and temperature compensation optical cable is used. The metal spiral armored optical cable in the three boreholes forms a loop at the bottom of the hole, and the transmission optical cable is connected to the transmission optical cable in one inlet and one outlet respectively to form a complete optical fiber monitoring system. It can not only describe the deformation information of the whole section of the borehole, but also realize the automatic monitoring in the field. The specific monitoring system layout process is as follows: After the completion of the on-site drilling, the sensing cable is placed in it, and a certain proportion of clay ball, fine sand and gravel mixture is backfilled according to the nature of the on-site stratum. During the backfilling process, the sensing cable is kept in a straight and pretensioned state to ensure that the fiber and the surrounding rock of the borehole are well coupled. Finally, the sensing signal is transmitted to the ground through the signal transmission cable, and the portable dense distributed optical fiber demodulator is used to demodulate the strain of the sensing cable. Through the analysis and processing of the strain data, the formation deformation information in the borehole is extracted.

The wavelength range of the demodulator used in the field test is 1528~1568 m, the wavelength resolution is 1 pm, and the repeatability is $\pm 2$ pm. Internal 4-channel design, single-channel demodulation rate < 5 s, single-channel measurement distance up to 20 km, the maximum number of measuring points 2000, all meet the test requirements.

The initial monitoring began at 21:37 on 15 November 2021. The longitudinal horizontal distance between the shield cutter head and the monitoring section was 30 m, and the shield was progressing at a rate of 4 rings/day (about 1.5 m per ring). The shield cutter head reached the bottom of the monitoring section at 21:30 on 20 November. At 21:25 on 21 November, the shield body was detached entirely (the longitudinal horizontal distance between the cutter head and the monitoring point was 6 m) and moved forward till it was connected.

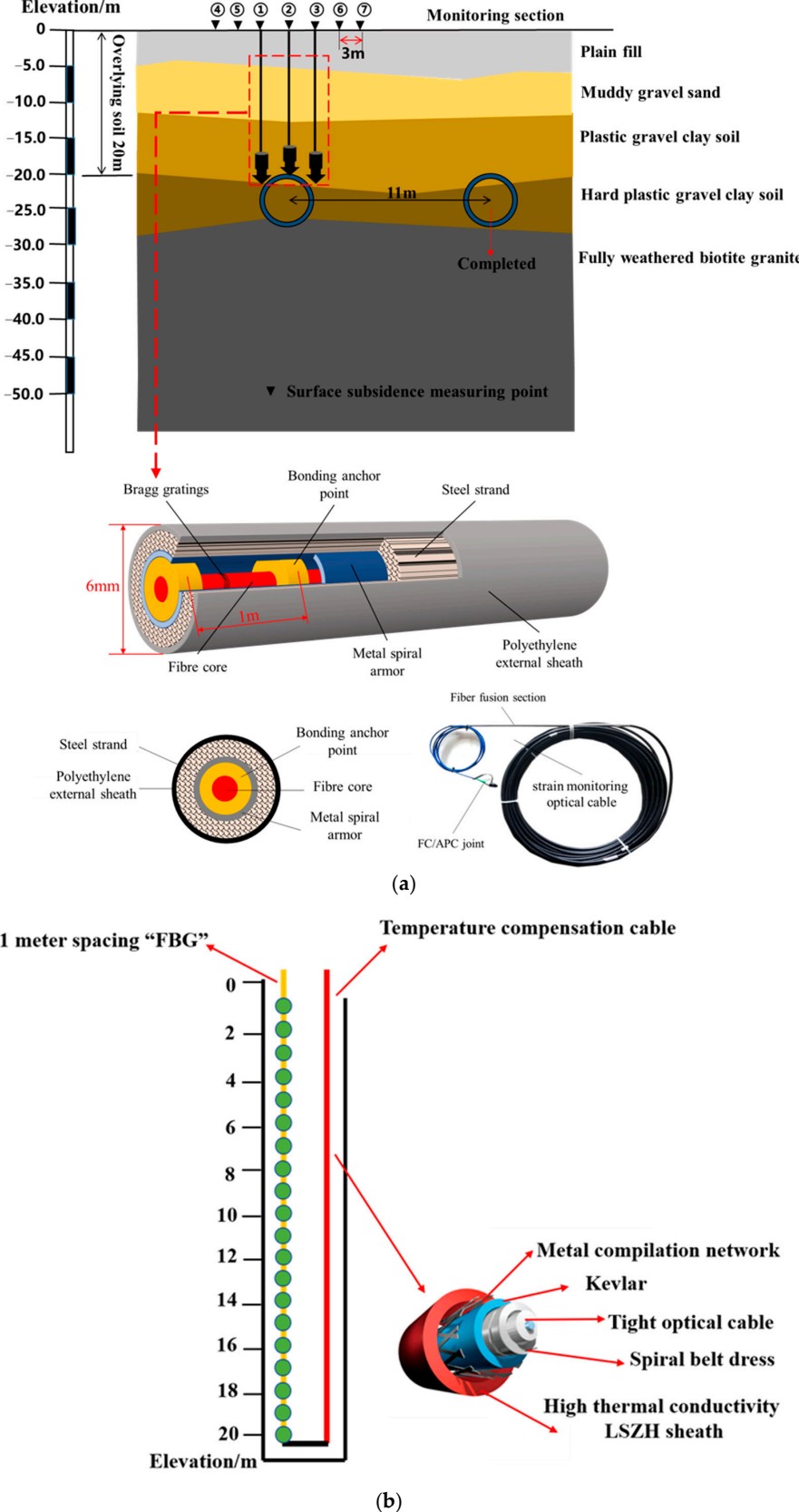

**Figure 2.** Layout of monitoring system. (**a**) Monitoring section layout. (**b**) Optical cable layout.

*2.3. Establishment of Numerical Simulation Model*

Based on site engineering survey and full consideration of tunnel excavation process stability, the calculation model was reasonably simplified, and the following basic assumptions were made in the finite element numerical simulation calculation: (1) it was assumed that the rock and soil mass (using the Mohr-Coulomb constitutive model) is homogeneous and isotropic in the same stratum; (2) the groundwater is in a lower position, without considering the seepage effect and the soil deformation's time effect, that is, without considering the secondary consolidation and creep effect of the stratum, and with assuming that the shield advance is a change in spatial displacement; (3) it was assumed that the shield advancing process is continuous, and the advancing length of each shield ring is 1.5 m; (4) in the calculation process, some parameters were adjusted appropriately to make the calculation more reasonable.

The three-dimensional numerical simulation modeling of the shield tunnel is carried out using ABAQUS nonlinear finite element analysis software. In this study, the stratum soil, shield, and lining segments were simulated using three-dimensional solid 8-node hexahedral linear reduced integration elements (C3D8R), making up a total number of 184,380 numerical model elements.

The numerical model was defined such that the x-axis is along the radial direction of the tunnel, the y-axis is along the tunnel's axial direction, and the z-axis is along the depth direction of the stratum. Among them, the maximum principal stress $\sigma_1$ of the tunnel with advancing TBM face is along the tunnel axis (y-axis), and the minimum principal stress $\sigma_3$ is along the horizontal direction (x-axis) perpendicular to the tunnel axis, which is orthogonal to the principal stress $\sigma_1$ in azimuth, as shown in Figure 3. Since the length of each ring lining segment of the shield tunnel project is 1.5 m, the actual project was advanced at a speed of 4 rings (6 m/d). In order to simulate the actual excavation process, the y-axis simulation range of the numerical model was set to be 60 m. The excavation area unit was divided into 20 sections during numerical modeling, with each section being 3 m long. Considering the influence of the boundary effect in the numerical modeling, the x-axis simulation range of the numerical model was set to 60 m, and the simulation range on both sides of the tunnel was set about 5 times the diameter of the tunnel. It can be considered that the boundary conditions on both sides of the model do not affect the simulation results [49–52]. The upper surface of the model was taken as the ground surface, the buried depth of the shield was set to 20 m, the length of the shield was defined as 6 m, the width of the lining segment was set to 1.5 m, the outer diameter was defined as 6.0 m, the inner diameter defined as 5.0 m, and the thickness was taken as 0.5 m. The z-axis simulation range of the numerical model was set to 50 m, and the influence of the bottom boundary was ignored. In the overall numerical model, the model dimensions are 60 m in length, 60 m in width, and 50 m in height, all of which have considered the boundary effects on the surrounding soil during tunnel excavation. Therefore, the upper surface of the model was set as a free boundary, horizontal constraints were applied to the front, rear, left and right sides. and vertical and horizontal constraints were applied to the lower surface of the model.

According to the actual engineering geological survey data on-site, the strata of the numerical simulation section were set from top to bottom as plain fill, muddy gravel sand, plastic gravel clay, hard plastic gravel clay, and completely weathered biotite granite, and the vertical distribution of each stratum was uneven. Each stratum soil's volume equivalent substitution method was appropriately treated in the model's longitudinal range. After simplification, the thickness of each stratum was set to 5.0 m for plain filling soil layer, 5.0 m for muddy gravel sand layer, 7.0 m for plastic gravel clay layer, 13.0 m for hard plastic gravel clay layer, and 20.0 m for completely weathered biotite granite layer. The basic physical and mechanical parameters of each stratum are shown in Table 1. The concrete grade of the lining segment structure was defined as C50, the material weight was taken as 25 kN/m, and the elastic modulus was defined as 35 GPa. Considering the influence of the circumferential joint's lining on the stiffness of the lining structure, the stiffness was

reduced by 0.2, and the stiffness reduction coefficient was taken as 0.8. The elastic modulus of the lining structure was defined as 28 GPa, and Poisson's ratio was taken as 0.2. The constitutive model adopted in this study is an isotropic elastic-plastic model that conforms to the Mohr-Coulomb yield criterion.

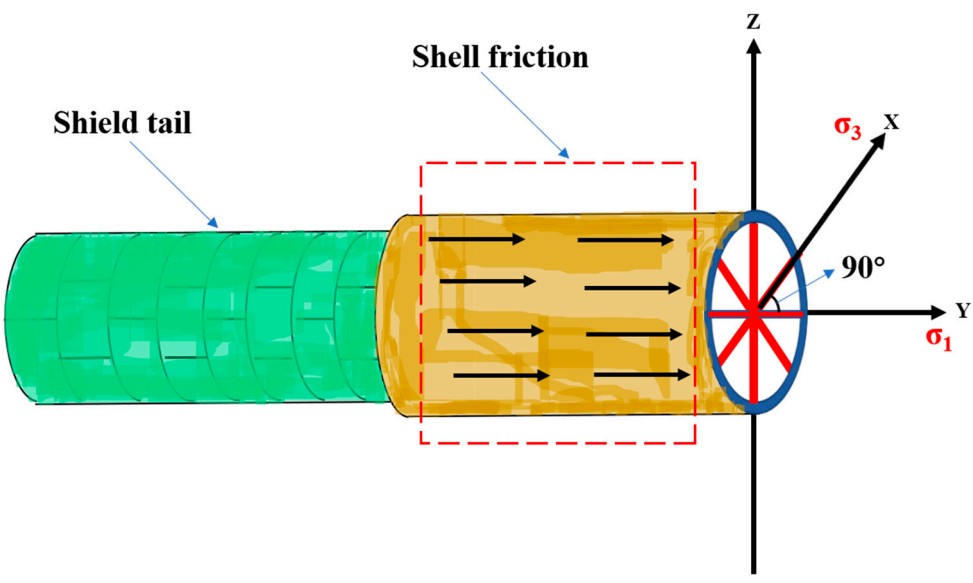

**Figure 3.** The directions of principle stresses around the tunnel with advancing TBM face.

**Table 1.** Basic mechanical indexes of soil in each stratum.

| Strata Name | Density (kN/m³) | Deformation Modulus (MPa) | Poisson's Ratio | Cohesion (kPa) | Friction Angle (°) | Strata Thickness (m) |
|---|---|---|---|---|---|---|
| Plain fill | 21.0 | 16 | 0.30 | 6 | 12 | 5 |
| Muddy gravel sand | 19.5 | 18 | 0.31 | 12 | 10 | 5 |
| Plastic gravel clay soil | 18.3 | 15 | 0.34 | 20 | 28 | 7 |
| Hard plastic gravel clay soil | 19.0 | 30 | 0.32 | 23 | 26 | 13 |
| Fully weathered biotite granite | 19.5 | 70 | 0.29 | 30 | 22 | 20 |

Shield construction mainly includes soil excavation and lining pipe assembly. The shield tunneling, soil excavation, and lining pipe assembly were simulated in ABAQUS using the raw and dead unit method. The total length of the excavated tunnel is 60 m, and each excavation step is 3 m, divided into 20 steps. The simulation steps in the specific construction process were as follows:

(1) The initial ground stress equilibrium was established, the initial ground stress field was obtained, and the initial displacement was returned to zero.

(2) ABAQUS birth and death element method was used to kill the 3 m excavation soil element, and the temperature field was used to change the soil elastic modulus and Poisson's ratio, which was used to simulate the stress release in the soil excavation process.

(3) The ABAQUS life and death element method activate the 3 m lining segment element in contact with the soil. Shield excavation of 3 m soil, assembling 3 m lining segments as a complete analysis step, the whole process of a total of 20 cycles to simulate the whole process of shield tunneling.

In the process of numerical simulation, the transverse monitoring section X was set up at y = 30 m along the tunneling direction of the shield tunnel, and 21 monitoring points were set up at the surface of the monitoring section. The spacing of each monitoring point is 3 m, symmetrically arranged on both sides of the tunnel axis, and the number is $A_1$–$A_{21}$ from left to right. Among them, the three monitoring points at the surface above the excavation boundary of the shield tunnel are $A_{10}$, $A_{11}$, and $A_{12}$. Surface monitoring points $A_{10}$, $A_{11}$, and $A_{12}$ from top to bottom, respectively, in the buried depth of 5 m, 10 m, 15 m, 17 m, and 20 m set strata subsidence monitoring points, in turn, numbered $B_{10}$–$F_{10}$, $B_{11}$–$F_{11}$, and $B_{12}$–$F_{12}$, as shown in Figure 4.

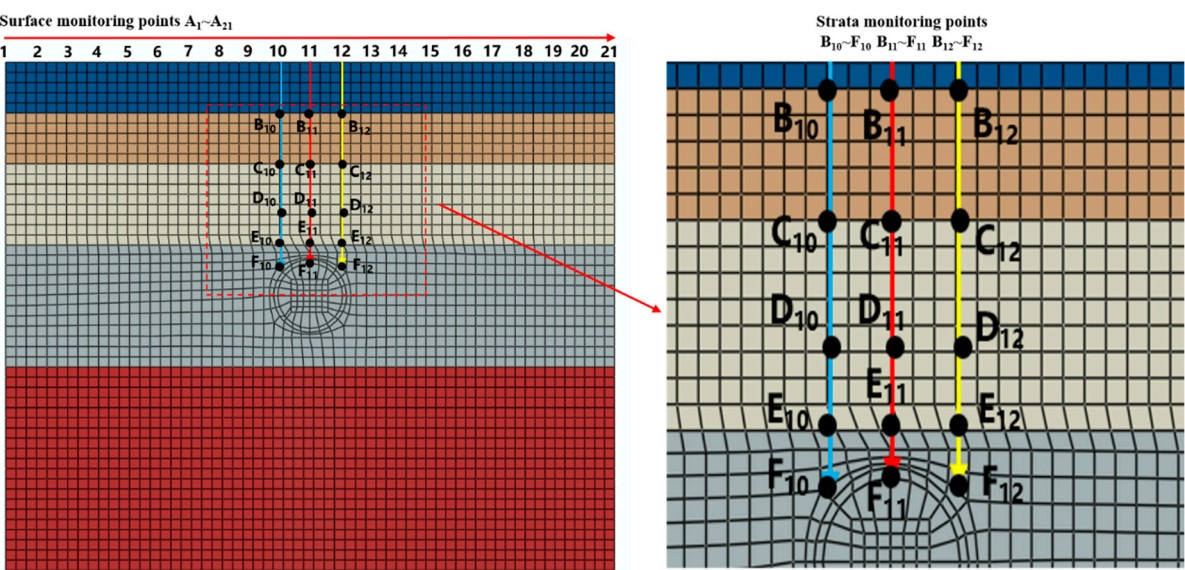

**Figure 4.** Numerical simulation monitoring point layout diagram.

## 3. Results and Analysis

By comparing the results of field monitoring and finite element analysis, the settlement law and distribution characteristics of composite stratum disturbed by shield tunneling are systematically analyzed. Mainly from the following two aspects: (1) The evolution law of surface subsidence; (2) Evolution law of stratum settlement at different depths with shield tunneling.

### 3.1. Evolution Law of Surface Subsidence

3.1.1. Field Monitoring Results

Figure 5 is the longitudinal settlement curve of holes 1 # through 3 # monitoring points on the surface. The longitudinal settlement curve of ground surface in the process of shield tunneling can be divided into three stages: shield approaching (−30~0 m), shield crossing (0~6 m) and shield tail falling out (6~30 m). Among them, the shield approaching stage can also be divided into initial deformation stage (−30~−6 m) and micro deformation stage (−6~0 m). The former curve changes gently, indicating that the shield tunneling disturbance has little effect on the surface settlement of the monitoring point. In the latter stage, the surface settlement deformation rate gradually increases but the deformation value is relatively small, and the maximum settlement of the approaching stage is reached when the cutterhead reaches the monitoring section. In addition, the surface subsidence increases rapidly from the shield crossing to the shield tail stripping stage, which is intuitively reflected in the increase of the slope of the curve. When the shield tail is separated from the cutterhead and passes through the monitoring section of 18 m (3D), the ground settlement reaches −11.59 mm, −12.04 mm and −11.19 mm, respectively. At this time, the settlement accounts for more than 2/3 of the total settlement, which is the main deformation zone disturbed by shield tunneling, and the subsequent displacement gradually stabilizes. Based

on the above analysis, it can be seen that the influence range of shield tunneling disturbance is about 3.0 times the hole diameter in front of the tunnel face to 1.0 times the hole diameter behind the tunnel face.

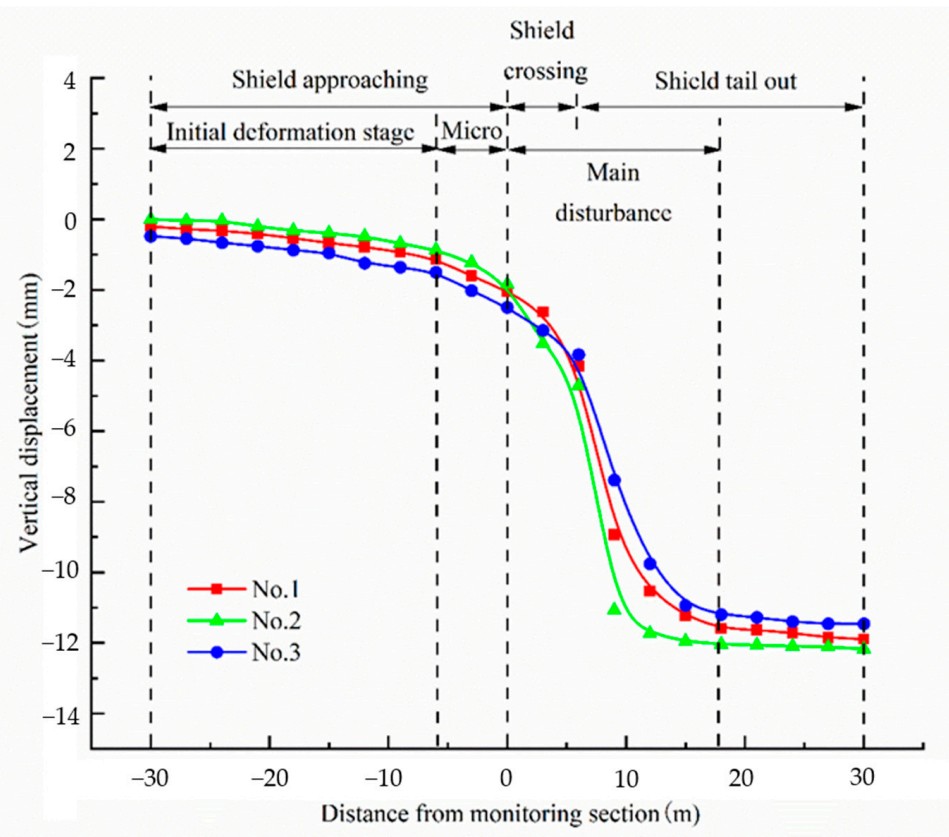

**Figure 5.** Variation of ground surface longitudinal settlement with the advancement of tunnel.

Figure 6 is the lateral displacement curve of the surface site monitoring point. It can be seen that when the shield cutter has just reached the monitoring section, the maximum surface settlement is −1.69 mm, the surface deformation is not obvious, there is no obvious settlement trough; when the cutterhead is 6 m away from the monitoring section, the maximum settlement value of the surface increases sharply, the maximum settlement value reaches −10.10 mm, and the settlement trough begins to appear gradually. When the shield cutterhead is 12 m away from the monitoring section, the lateral settlement of the surface tends to be stable, and the maximum settlement value reaches −11.59 mm. At this time, the shield cutterhead is about twice the outer diameter of the shield from the monitoring section, and the settlement trough is basically formed. In the stage of shield tail disengaging, the increasing amplitude of surface transverse settlement decreases gradually. When the shield cutterhead is 30m away from the monitoring section, the settlement value of transverse monitoring section reaches the peak value −12.77 mm. In addition, in the stage of shield crossing and shield tail disengagement, the surface settlement above the tunnel vault (hole 2 # surface monitoring point) is always the largest. The surface settlement is the most obvious in the range of 1 time the outer diameter of the shield (−6~6 m) on both sides of the tunnel center, and the influence range of the whole excavation disturbance is 1.5 times the outer diameter of the shield on both sides of the tunnel center (−9~9 m).

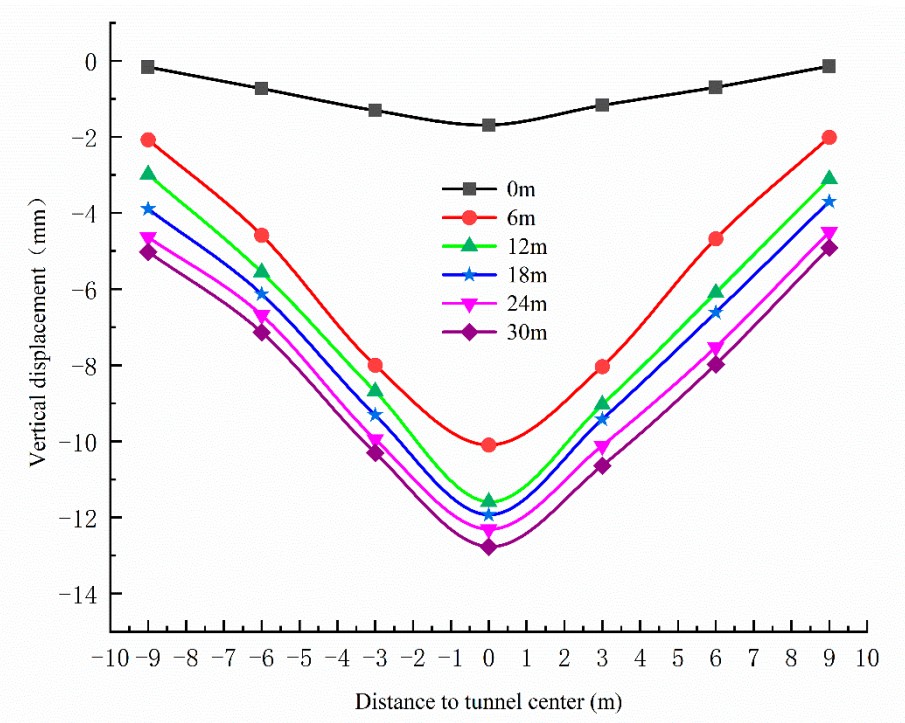

**Figure 6.** Transverse vertical displacement of the ground surface.

3.1.2. Numerical Simulation Results

In order to facilitate the intuitive study of the evolution law of surface settlement during shield tunneling, the vertical displacement cloud map of the monitoring section is processed by view slicing, as shown in Figure 7. It can be seen from the diagram that when the shield is close to the monitoring section of $-24 \sim -12$ m, the maximum surface settlement is 0.61 mm; in the 8th excavation step ($-6$ m from the monitoring section), the ground settlement is about 1.16 mm, and the 10th excavation step (0 m from the monitoring section) reaches 1.94 mm. Analysis shows that shield approaching stage with the cutterhead gradually close to the monitoring section, the surface subsidence value increases gradually but the deformation value is relatively small, the cutterhead from the monitoring section $-6 \sim 0$ m surface disturbance is more obvious. After the shield crosses the monitoring section, the ground settlement increases gradually and tends to be stable after reaching the maximum value of 12.04 mm in the 16th excavation step (18 m from the monitoring section).

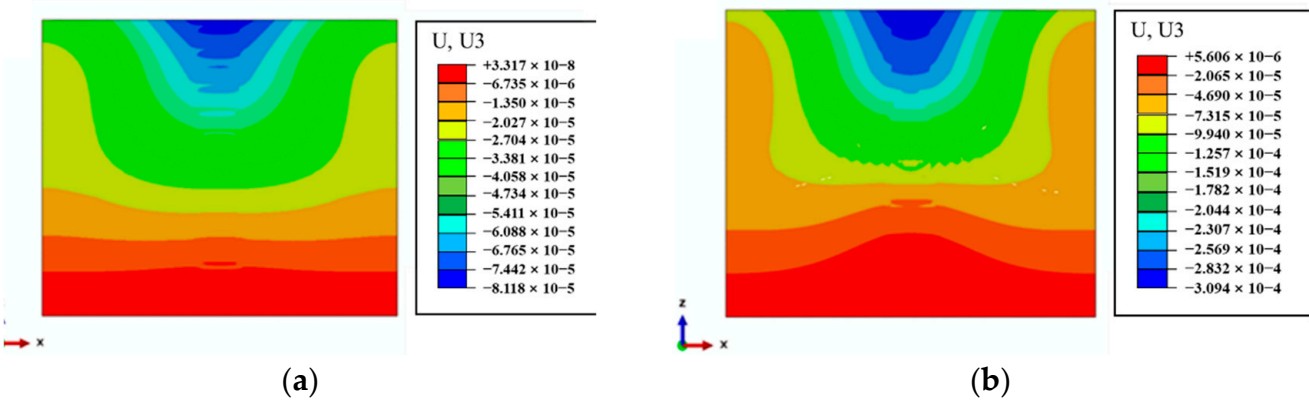

(a)　　　　　　　　　　　　　　　　　(b)

**Figure 7.** *Cont.*

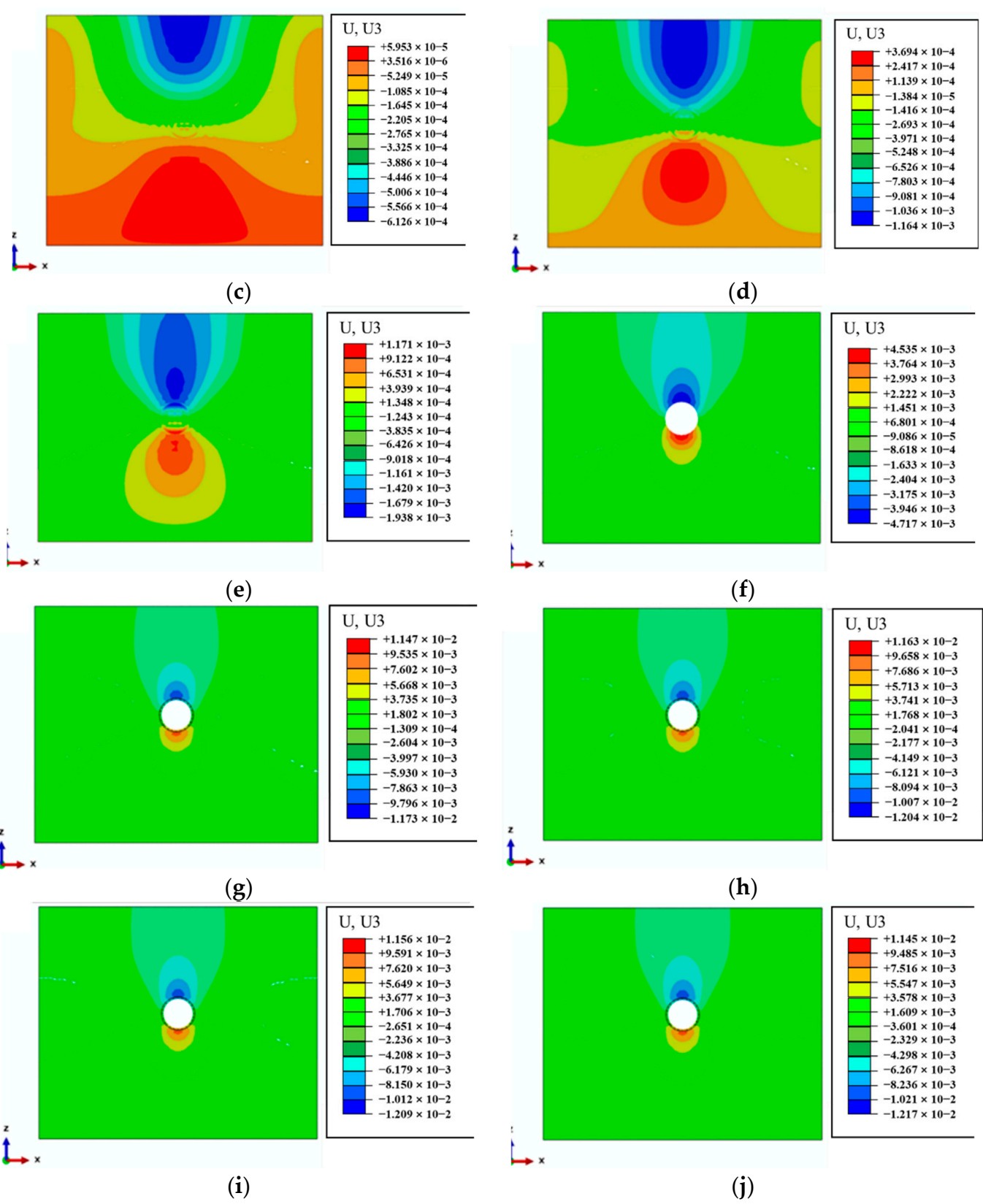

**Figure 7.** The cloud atlas of vertical displacement during different tunneling stages. (**a**) −24 m. (**b**) −18 m. (**c**) −12 m. (**d**) −6 m. (**e**) 0 m. (**f**) 6 m. (**g**) 12 m. (**h**) 18 m. (**i**) 24 m. (**j**) 30 m.

Tables 2–4 are the longitudinal displacement statistics of numerical simulation monitoring points $A_{10}$, $A_{11}$, $A_{12}$ and field monitoring holes 1 # through 3 # at different stages of shield tunneling. Figure 8 is the comparison curve of longitudinal displacement between field monitoring point and corresponding simulated monitoring point. It can be seen from the comparative analysis that the measured values of surface subsidence in the field are slightly larger than the simulated values, but the relative error is within 10%, which proves the correctness of the material parameters of the numerical simulation model and the applicability to the subsequent research.

**Table 2.** Longitudinal surface displacements of $A_{10}$ and hole 1 #.

| Distance (m) | −24 | −18 | −12 | −6 | 0 | 6 | 12 | 18 | 24 | 30 |
|---|---|---|---|---|---|---|---|---|---|---|
| Field monitoring (mm) | −0.32 | −0.84 | −1.20 | −1.62 | −2.37 | −4.15 | −10.53 | −11.59 | −11.72 | −11.89 |
| Numerical simulation (mm) | −0.29 | −0.78 | −1.08 | −1.51 | −2.23 | −3.97 | −10.14 | −11.07 | −11.31 | −11.53 |
| Absolute error (mm) | 0.03 | 0.06 | 0.12 | 0.11 | 0.14 | 0.18 | 0.39 | 0.52 | 0.41 | 0.36 |
| Relative error | 10.34% | 7.69% | 11.11% | 7.28% | 6.28% | 4.53% | 3.85% | 4.70% | 3.63% | 3.12% |

**Table 3.** Longitudinal surface displacements of $A_{11}$ and hole 2 #.

| Distance (m) | −24 | −18 | −12 | −6 | 0 | 6 | 12 | 18 | 24 | 30 |
|---|---|---|---|---|---|---|---|---|---|---|
| Field monitoring (mm) | −0.14 | −0.44 | −0.51 | −1.16 | −1.43 | −4.71 | −12.04 | −12.91 | −12.92 | −12.97 |
| Numerical simulation (mm) | −0.13 | −0.40 | −0.50 | −1.09 | −1.37 | −4.56 | −11.55 | −12.01 | −12.09 | −12.17 |
| Absolute error (mm) | 0.01 | 0.04 | 0.01 | 0.07 | 0.06 | 0.15 | 0.49 | 0.90 | 0.83 | 0.80 |
| Relative error | 7.7% | 1.0% | 2.0% | 6.4% | 4.4% | 3.3% | 4.2% | 7.8% | 6.9% | 6.6% |

**Table 4.** Longitudinal surface displacements of $A_{12}$ and hole 3 #.

| Distance (m) | −24 | −18 | −12 | −6 | 0 | 6 | 12 | 18 | 24 | 30 |
|---|---|---|---|---|---|---|---|---|---|---|
| Field monitoring (mm) | −0.74 | −1.23 | −1.81 | −2.39 | −3.09 | −3.73 | −9.75 | −11.19 | −11.40 | −11.46 |
| Numerical simulation (mm) | −0.69 | −1.18 | −1.75 | −2.19 | −2.89 | −3.58 | −9.46 | −10.29 | −10.39 | −11.04 |
| Absolute error (mm) | 0.05 | 0.05 | 0.06 | 0.20 | 0.20 | 0.15 | 0.29 | 0.90 | 1.01 | 0.42 |
| Relative error | 7.2% | 4.2% | 3.4% | 9.1% | 6.9% | 4.2% | 3.1% | 8.7% | 9.7% | 3.8% |

*3.2. Evolution Law of Stratum Settlement at Different Depths with Shield Tunneling*

3.2.1. Field Monitoring Results

Figure 9 is the field monitoring time history curve of layered settlement of hole 1 # through 3 # strata. It can be seen from the analysis that the overall change trend of the layered settlement law of each hole is roughly the same, and similar to the surface settlement law. Among them, the layered settlement of each stratum is not obvious in shield approaching stage; After the shield body passes through and the shield tail is pulled out, the settlement stratification of each stratum in the range of about 3 times the outer diameter of the shield (0~18 m) is obvious, and the settlement rate is significantly accelerated.

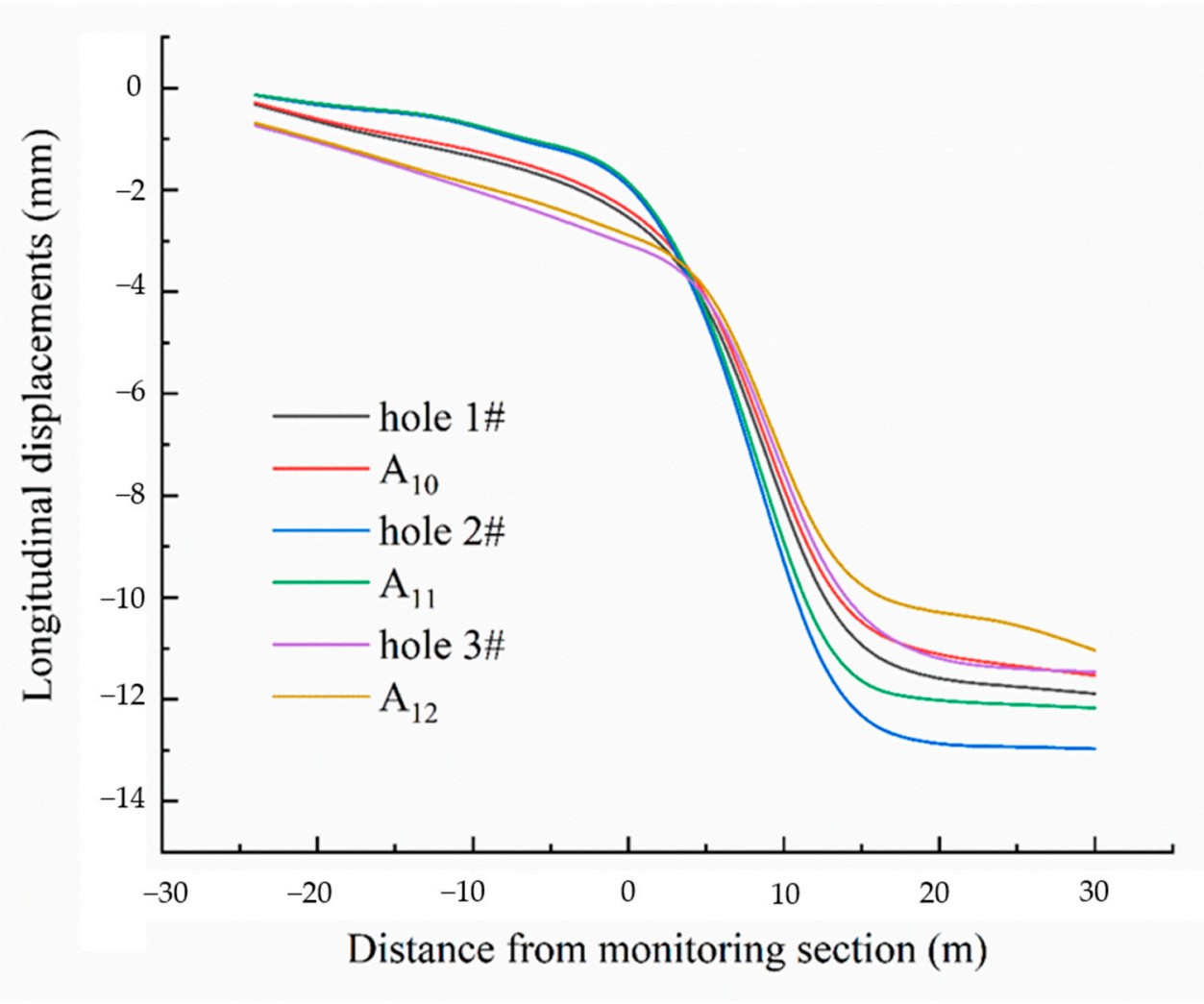

**Figure 8.** Longitudinal displacement comparison diagram.

From the horizontal point of view, 10~15 m away from the tunnel vault is muddy gravel sand stratum, which has the characteristics of high natural water content, large void ratio, high compressibility, long consolidation time and high sensitivity, and is more susceptible to shield tunneling disturbance, showing more obvious settlement deformation. At the same time, there are significant peak differences and time asynchrony characteristics in this area. It is shown that the hole 1 # and hole 3 # reach the peak settlement of 2.21 mm and 1.72 mm respectively when the cutter head is 12 m away from the monitoring section, while the hole 2 # reaches the peak settlement of 4.41 mm when the cutter head is 18 m away from the monitoring section. The reason for this phenomenon is that the ground disturbance above the shield is more severe and lasts longer. In addition, 0.5 times the outer diameter of the shield (0~3 m) above the tunnel vault also produces large settlement deformation. The settlement peak is 2.42 mm, accounting for 21.1% of the total settlement, and is different from the hole 1 # and hole 3 #. The hole 2 # stratum shows the characteristics of large layered settlement amplitude and wide longitudinal influence range.



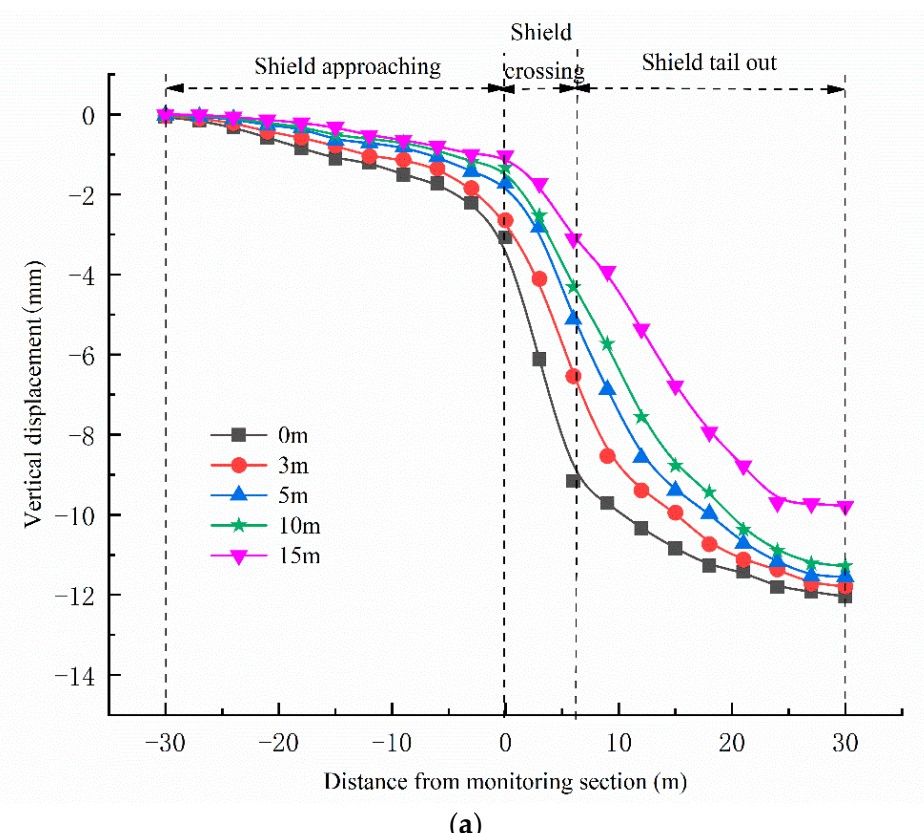

(**a**)

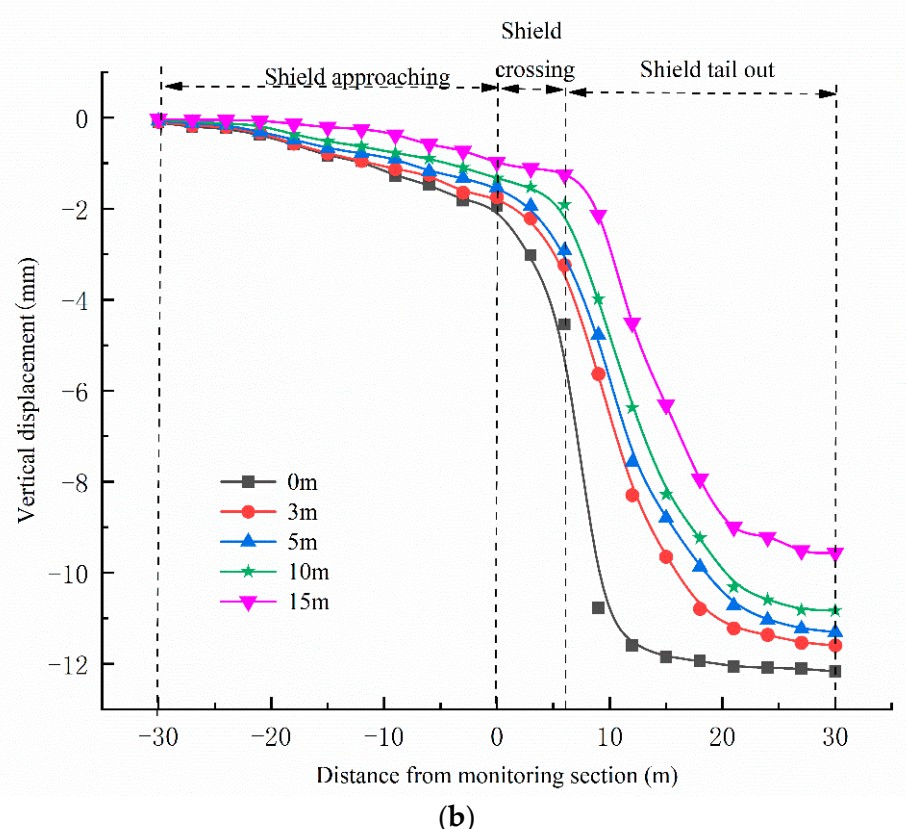

(**b**)

**Figure 9.** *Cont.*

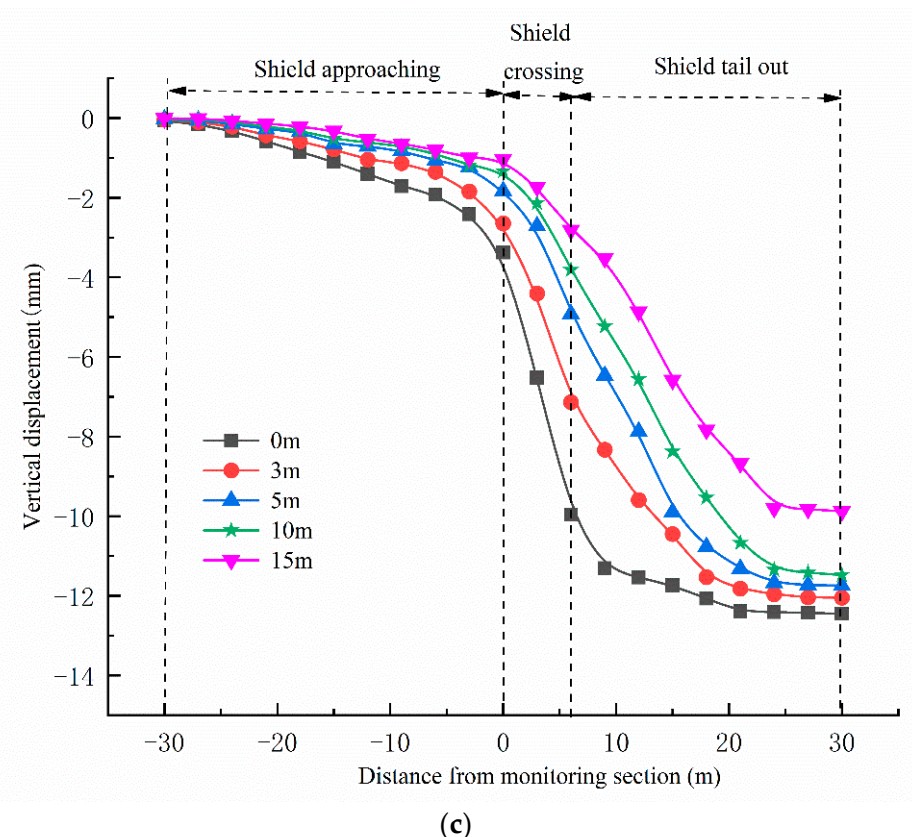

**Figure 9.** Layered subsidence time history curve of monitoring section stratum. (**a**) hole 1 #. (**b**) hole 2 #. (**c**) hole 3 #.

### 3.2.2. Numerical Simulation Results

It can be seen from the above field monitoring analysis that the strata within the range of 0.5 times the outer diameter of the shield (0~3 m) above the tunnel produce large settlement deformation. In order to intuitively reflect the settlement law, the numerical simulation results of stratum settlement at 3 m above the tunnel are compared with the field measured values, as shown in Figure 10. Tables 5–7 show the variation of the proportion of stratum settlement at 3 m above holes 1 # through 3 # tunnel in total settlement with different stages of shield tunneling.

Combined with the chart analysis, it can be seen that the deviation between the field monitoring value and the numerical simulation result is small, the overall change trend is basically consistent, and the settlement in the shield tail stripping stage accounts for the largest proportion. Among them, the settlement of shield approaching stage (−30~0 m) is small. Taking the hole 2 # with the largest settlement proportion in this stage as an example, its settlement accounts for 22.8% of the total settlement, while the corresponding numerical simulation settlement accounts for about 11.0% of the total settlement. With the continuous advancement of tunnel excavation, the settlement deformation rate continues to increase, and the settlement proportion of shield crossing stage (0~6 m) continues to increase. The field measurement shows that this stage accounts for 27.9%, 28.4% and 38.0% of the three holes respectively, reaching about 1/3 of the total settlement. At this stage, the proportion of settlement based on numerical simulation also continued to increase, but the overall increase rate was low. The reason for this difference may be that the stratum was disturbed for a longer time during the actual construction process. Based on the field monitoring and numerical simulation results, the proportion of settlement is greater than 40% in the shield tail detachment stage (6~30 m). At this stage, the stratum settlement increases significantly and is the main settlement stage.

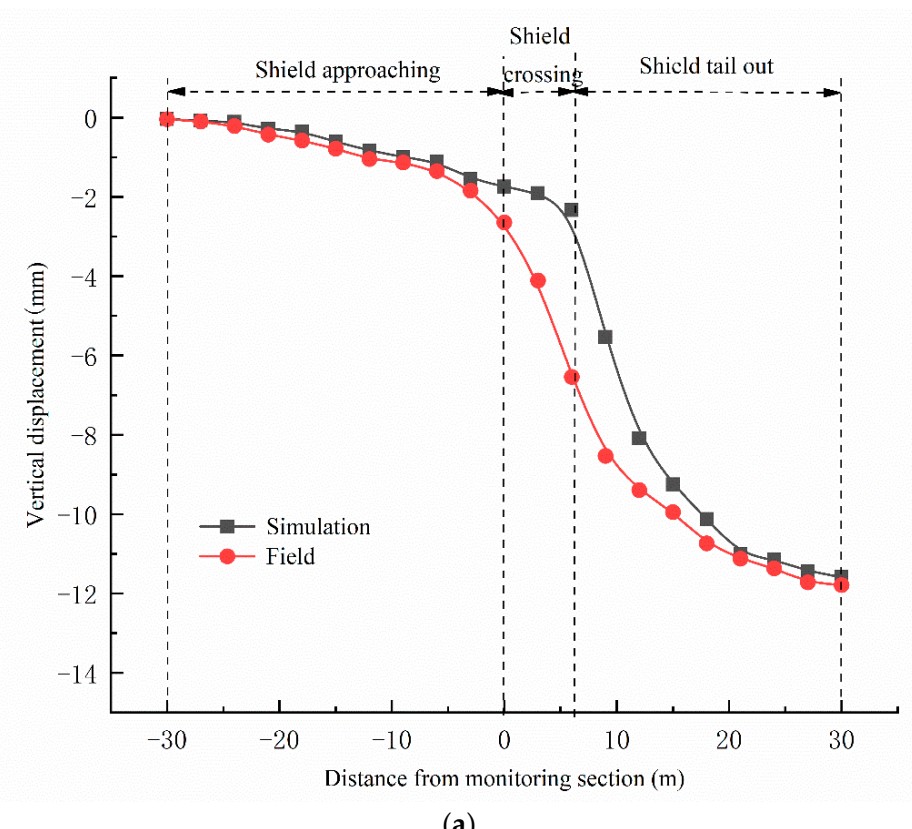

(**a**)

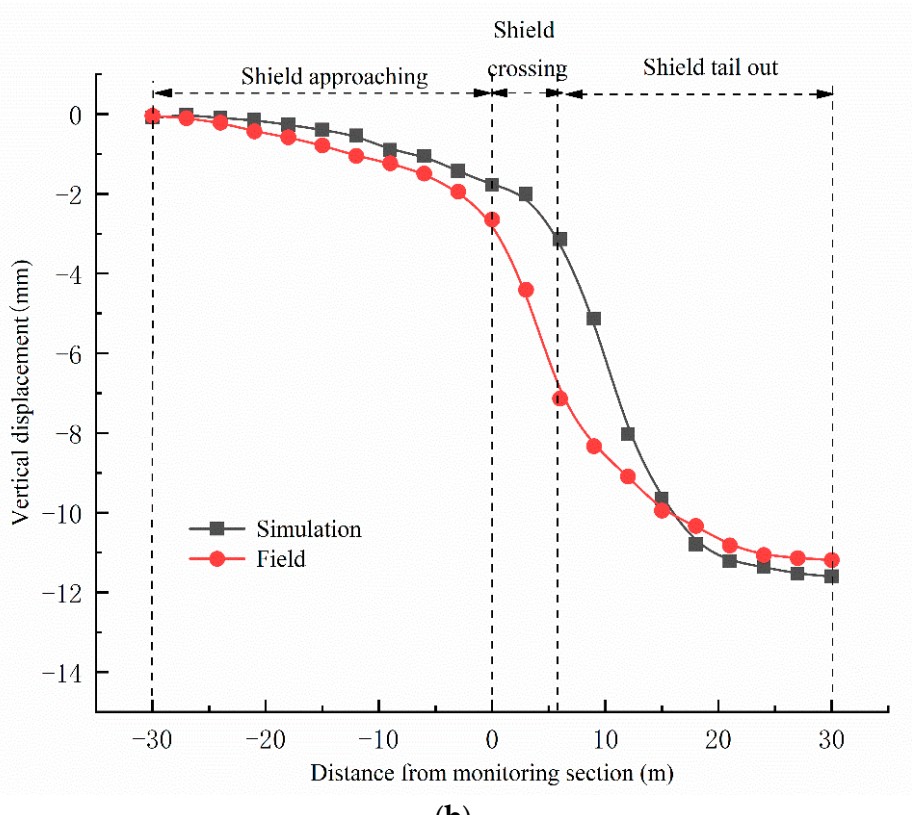

(**b**)

**Figure 10.** *Cont.*

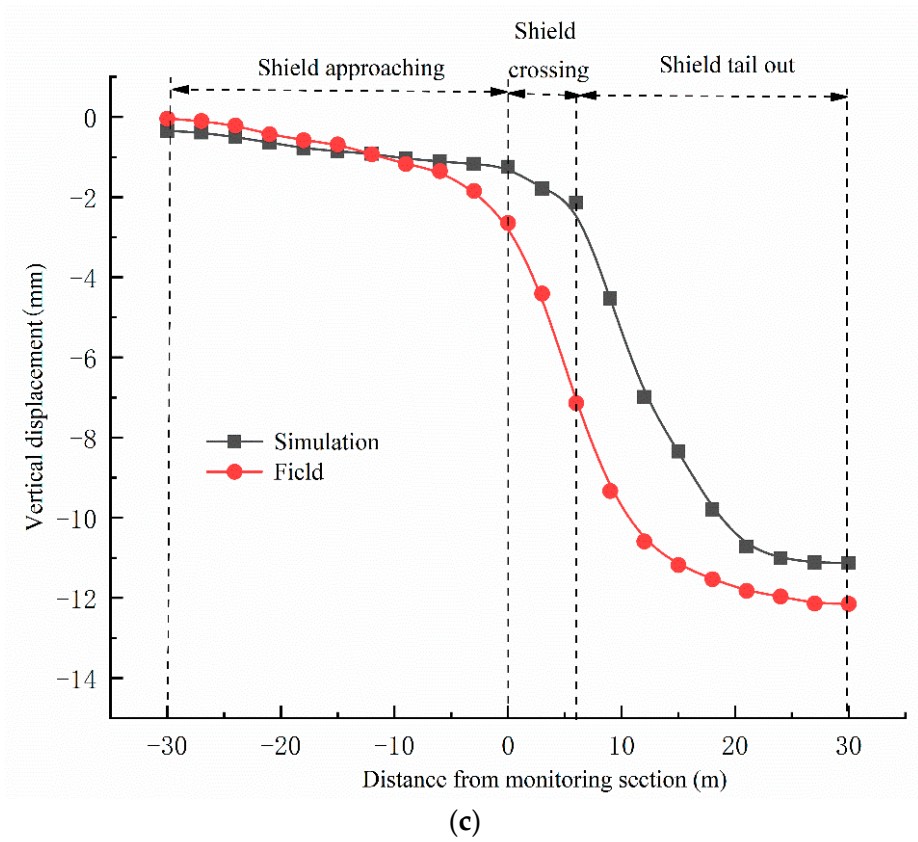

(**c**)

**Figure 10.** Layered settlement time history curve of stratum at 3m above the tunnel. (**a**) hole 1 #. (**b**) hole 2 #. (**c**) hole 3 #.

**Table 5.** The stratum settlement value at 3 m above the tunnel at different stages of shield tunneling in hole 1 #.

| Transverse Distance (m) | −30~0 | 0~6 | 6~30 | Total |
|---|---|---|---|---|
| Field monitoring (mm) | −2.65 | −3.29 | −5.84 | −11.78 |
| Ratio | 22.5% | 27.9% | 49.6% | / |
| Numerical simulation (mm) | −1.14 | −1.20 | −9.20 | −11.54 |
| Ratio | 9.9% | 10.4% | 79.7% | / |

**Table 6.** The stratum settlement value at 3 m above the tunnel at different stages of shield tunneling in hole 2 #.

| Transverse Distance (m) | −30~0 | 0~6 | 6~30 | Total |
|---|---|---|---|---|
| Field monitoring (mm) | −2.73 | −3.41 | −5.85 | −11.99 |
| Ratio | 22.8% | 28.4% | 48.8% | / |
| Numerical simulation (mm) | −1.28 | −1.86 | −8.46 | −11.60 |
| Ratio | 11.0% | 16.0% | 73.0% | / |

**Table 7.** The stratum settlement value at 3 m above the tunnel at different stages of shield tunneling in hole 3 #.

| Transverse Distance (m) | −30~0 | 0~6 | 6~30 | Total |
|---|---|---|---|---|
| Field monitoring (mm) | −2.50 | −4.62 | −5.03 | −12.15 |
| Ratio | 20.6% | 38.0% | 44.4% | / |
| Numerical simulation (mm) | −1.24 | −1.68 | −8.21 | −11.13 |
| Ratio | 11.1% | 15.1% | 73.8% | / |

## 4. Discussion

Based on the comparative study of numerical simulation and weak reflectivity fiber grating field monitoring, the evolution law and distribution characteristics of vertical and horizontal settlement of shield tunneling composite strata are comprehensively analyzed. On this basis, according to the influence area of surrounding buildings (structures), targeted control measures are put forward.

From the longitudinal perspective, the temporal and spatial evolution law of layered settlement of composite stratum with shield tunneling can be divided into three stages according to different stages of shield tunneling. (1) Shield approaching stage: The angle between the maximum principal stress axis and the x-axis remains basically unchanged. The angle between the maximum principal stress axis and the y and z-axes continues to increase, but the amplitude is small and the frequency is slow, the direction of the maximum principal stress is mainly rotated along the vertical plane parallel to the tunnel axis. A small settlement is caused by the influence of shield thrust, ranging from the cutter head ($-5D$~$0$); (2) Shield crossing stage: Affected by factors such as excavation gap and continuous disturbance, the maximum principal stress increases sharply, but the angle with the y-axis is basically unchanged, the minimum principal stress continues to increase, and the direction also gradually changes, the stratum settlement gradually increases, and the range is from the cutterhead ($0$~$1D$); (3) Shield tail out stage: The direction of the minimum principal stress changes obviously. The angle between the minimum principal stress and the x and y-axes increases continuously, and the angle between the minimum principal stress and the z axis increases slowly. It mainly rotates in the horizontal plane along the tunnel axis, rapid increase in formation settlement, later settlement rate gradually decreased and stabilized, mainly by the creep of soil or compression consolidation, range from the cutterhead ($1D$~$5D$). In addition, the longitudinal main disturbance area of layered settlement is about 3D range after shield crossing and shield tail stripping [53–56], as shown in Figure 11.

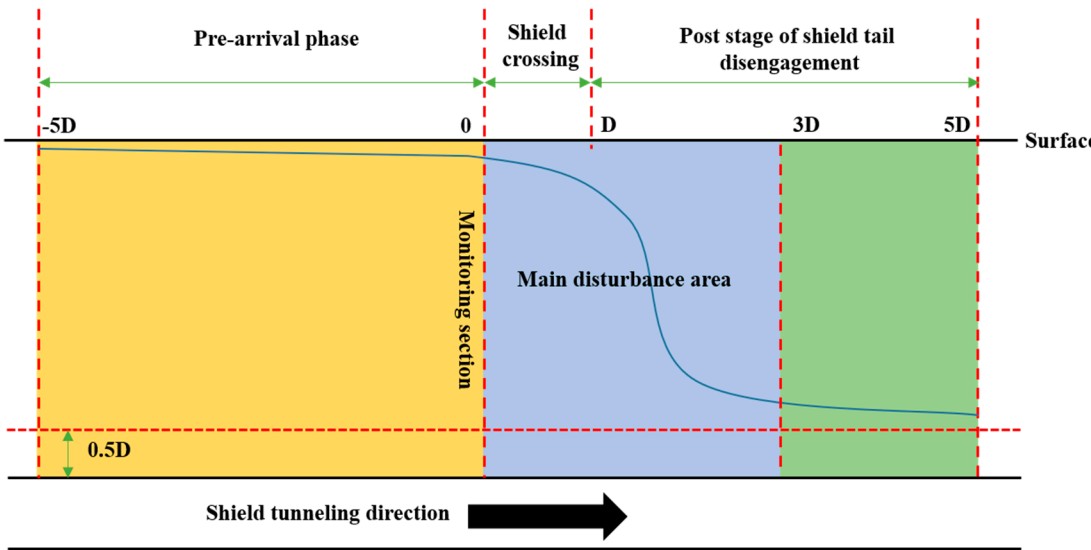

**Figure 11.** Vertical disturbance area division diagram.

From the horizontal perspective, according to the influence degree of shield tunneling disturbance, the composite stratum above the tunnel can be roughly divided into the main disturbance layer and the secondary disturbance layer. Among them, the main disturbance layer is located in the range of 0.5D above the tunnel, and its settlement accounts for about 80% of the total settlement of the composite strata, as shown in Figure 12.

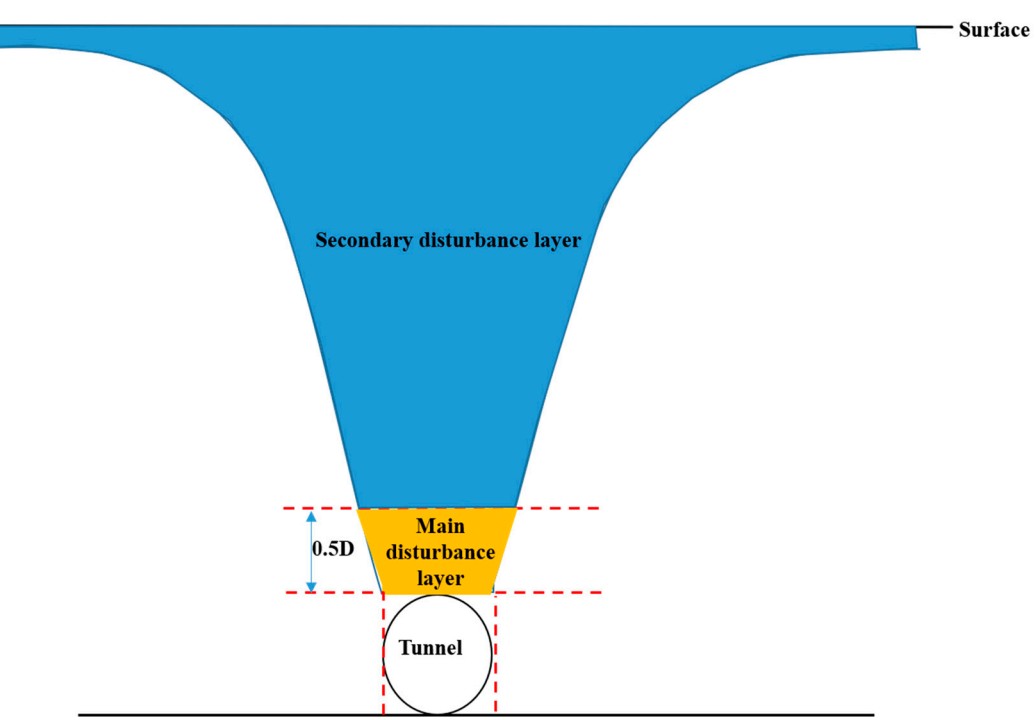

**Figure 12.** Horizontal disturbance layer division diagram.

Based on the vertical and horizontal zoning (layers) of existing structures such as buildings (structures) at different stages of shield tunneling, the following layered settlement control measures and suggestions for composite strata are proposed:

(1) For the existing buildings (structures) located in the main disturbance layer and the main disturbance area, the above research shows that the soil unloading in this area is obvious in the shield approaching stage. R. J. Finno and G. W. Clough [57] through the field measurement and finite element simulation of the shield construction of the San Francisco tunnel in the United States, it is found that properly increasing the cutterhead pressure to make the ground in front of the excavation face slightly uplift in advance can reduce the total ground loss during the construction period, and then better control the settlement during the construction process. This experience is confirmed by most projects. Therefore, the support pressure of excavation face should be appropriately increased to balance the overload effect and reduce the settlement caused by unloading of excavation face before crossing; During the shield crossing stage, the shield tunneling speed should be kept stable to reduce the adverse effects on the surrounding strata and existing buildings. Properly increasing the shield tunneling speed can reduce the amount and development speed of ground settlement. The shield advance speed of this project is controlled at 25~30 mm / min; The shield tail stripping stage is the main stage of disturbance. Synchronous grouting should be used to fill the excavation gap, and secondary grouting should be used to fill the gap behind the segment. Colleagues should avoid abnormal shutdown of shield at this stage, and strive to minimize the time of stripping stage.

(2) For the existing buildings (structures) located in the main disturbance layer and the secondary disturbance area, the shield crossing stage, to ensure that the shield uniform construction, shield attitude change should not be too large, over-frequency. In this project, the advance of the segment is checked every 4 rings, and the change of the folding angle between the tunnel axis and the shield axis cannot exceed 0.4 %. To avoid the excessive angle between the shield and the segment, the plane position of the shield machine is controlled within the design axis ±50 mm, and the elevation is controlled within −50 mm. At the same time, in order to reduce land subsidence, in

the process of crossing, it is strictly prohibited to correct a large number of deviations, only less or no correction.

(3) For the existing buildings (structures) located in the secondary disturbance layer and the main disturbance area, the tunneling speed, tunneling attitude, cutterhead torque and rotational speed should be paid attention to in the approaching stage of shield tunneling. Under the premise of ensuring the smooth tunneling of the excavation face, the cutterhead speed should be increased and the cutterhead torque should be reduced, which is conducive to the control of stratum settlement; During the shield crossing stage, the long time shelving of the shield machine should be avoided and the crossing interval should be minimized on the basis of the above basic control measures. In the stage of shield tail detachment, synchronous grouting is needed to fill the gap of shield tail, and then whether secondary grouting is needed is determined according to the real-time monitoring data of stratum settlement.

(4) For the existing buildings (structures) located in the secondary disturbance layer and secondary disturbance area, the tunneling speed and attitude should be paid attention to in the approaching and crossing stage of shield tunneling; In the stage of shield tail detachment, synchronous grouting was used to fill the shield tail gap, and then according to the real-time monitoring data of stratum settlement, whether to use secondary grouting to fill the gap behind the segment was determined.

## 5. Conclusions

This paper takes the shield project of "Keyuan Station ~ Shenzhen University Station" section of Shenzhen Metro Line 13. Through the combination of theoretical analysis, numerical simulation and field test, the layered settlement characteristics of shield tunnel crossing composite strata are systematically analyzed. The following main conclusions are drawn:

(1) The weak reflectivity fiber grating sensing technology can better perceive the evolution law and distribution characteristics of vertical and horizontal settlement of composite strata caused by shield tunneling, which is in good agreement with the numerical simulation results, and has the advantages of automation and high precision, it can be used as a supplement and alternative method for traditional measurement methods.

(2) The vertical and horizontal partition (layer) system of layered settlement of composite strata with the temporal and spatial evolution of shield tunneling is constructed. The temporal and spatial evolution law of ground settlement at different depths with shield tunneling can be divided into three stages in the longitudinal direction, namely, the shield approaching stage (−5D~0), the shield crossing stage (0~1D) and the shield tail detachment stage (1D~5D), and the 3D range after the shield crossing and the shield tail detachment is the longitudinal main disturbance area of layered settlement. Horizontally, the overlying composite strata are divided into main disturbance layer and secondary disturbance layer according to the influence of shield tunneling disturbance. Among them, the range of 0.5D stratum above the tunnel is the main disturbance layer.

(3) According to the influence zone of the building (structure) and the different stages of shield construction, corresponding effective control measures can be taken to achieve accurate control of stratum displacement and safe and efficient tunneling of shield.

**Author Contributions:** Conceptualization, F.Z. and X.L.; methodology, F.Z. and X.L.; validation, F.Z. and H.S.; formal analysis, X.L. and B.L.; investigation, S.L.; resources, S.L. and K.D.; data curation, Y.F.; writing—original draft preparation, F.Z.; writing—review and editing, F.Z. and X.L.; funding acquisition, H.S. All authors have read and agreed to the published version of the manuscript.

**Funding:** The authors would like to acknowledge the support provided by the 2019 China Construction Technology Research and Development Project Research on Key Technologies of Subway Construction under Multi-Factor Coupling (CSCEC-2019-Z-19).

**Institutional Review Board Statement:** The participant's personal identification information used in the study did not include personal information. Ethical review and approval were not required for the study.

**Informed Consent Statement:** Not applicable.

**Data Availability Statement:** Data sharing is not applicable.

**Acknowledgments:** The authors would like to thank the anonymous reviewers for their valuable comments that helped improve the paper's quality. We also thank China Construction Communications Engineering Group Co., Ltd. for helping us collect research data.

**Conflicts of Interest:** The authors declare no conflict of interest.

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
