# Peer review of "Study on Stratified Settlement and Weak Reflectivity Fiber Grating Monitoring of Shield Tunnel Crossing Composite Strata"

_applsci, doi:10.3390/app13031769_

Round 1

Reviewer 1 Report

Journal: Applied Sciences (ISSN 2076-3417)

Manuscript ID: applsci-2094476

Title: Study on Stratified Settlement and Weak Reflectivity Fiber Grating Monitoring of Shield Tunnel Crossing Composite Strata

Authors: Fucai Zhao , Xingli Lu * , Hongbing Shi , Bin Liu , Shaoran Liu , Kaohong Dai , Ying Fan

In this manuscript, the authors presented theoretical analysis, finite element analysis and field tests project to study the layered settlement characteristics of shield tunnel crossing composite strata.

The authors addressed the knowledge gaps through literature review. The content is important for practical application. Therefore, it is recommended to accept the manuscript.

Author Response

Thanks for your suggestion. Please see the attachment.

Reviewer 2 Report

1. Some paragraphs are aligned at both ends, such as lines 231-238; Some paragraphs are indented before the body, suggesting consistent adjustments by the writer.

2. The length of abstract is too long. It is suggested that the author make some cuts in the abstract.

3. Project overview: Add the scale in Figure 1, which is absolutely necessary.

4. Proofreading in English is recommended to eliminate some grammatical errors in the paper.

5. Line 198, "Can be considered......" Some relevant literatures can be cited for verification

6. It is necessary to summarize paragraph (1), (2) and (3) of Section 4.

Author Response

(The authors gave the same response as above.)

Reviewer 3 Report

The presented paper describes the use of weak optical fiber sensing for the monitoring of vertical deformation in soil layers over an advancing tunnel line. These measurements were then validated through numerical modelling.

Although well-written, there are no new insights proposed in the manuscript. There is no gap identified for which the presented study addresses, and no substantial novelty identified. The numerical model serves no further function than as a 'curve fitting' exercise. There is an absence of details pertaining to the development of the adopted constitutive models and values. The proposed main and secondary disturbance layers could be described using other existing theories, e.g. soil arching theory.

For these reasons, this Reviewer recommends for the manuscript to be rejected.

Author Response

(The authors gave the same response as above.)

Reviewer 4 Report

Reviewer’s comments

1.       The abstract is too long. The most important findings should be highlighted.

2.       Can you explain what you mean by shield tunnel construction?

3.       In Figure 1, position (a) beneath the top graph.

4.       Explain why the stratum settling law during the penetration of the left line shield tunnel was only addressed in this study and the effect of the right line shield tunnel was ignored.

5.       Fig. 2 (a) illustrates the tunnel only passing through hard plastic gravel clay soil and fully weathered biotite granite, whereas Fig. 1 portrays the tunnel passing through plastic, hard plastic gravel clay soil, and fully weathered biotite granite.

6.       Does the backfill material (e.g., clay ball, fine sand, and gravel combination) affect the temperature and transmission of the sensor cables?

7.       It is advised that the results of longitudinal displacements in Tables 2-4 be concluded by presenting a graph that compares all field monitored results to numerical ones for the three holes #1, #2, and #3, as well as their associated surface points #A10, #A11, and #A12.

8.       Lines 477-479 in the discussion section are similar to lines 481-482; delete the latter (e.g., Lines 481-482). Similarly, Lines 479-481 and Lines 482-484; delete Lines 479-481.

Author Response

(The authors gave the same response as above.)

Round 2

Reviewer 3 Report

The Reviewer thanks the Authors for providing an explanation. Some suggestions are made for the Authors' considerations:

Lines 67 - 69: Please include references for each of the methods stated, i.e. "theoretical analysis, model tests, field tests and other methods".

Under the Introduction, please reference articles which make use of RFB. Identify the gap and / or novelty of using RFB in tunnel monitoring.

In the numerical model, please show the directions of principle stresses around the tunnel with advancing TBM face. Please consider this for the discussion of results, and for Figure 11. If relevant, please refer to articles on soil arching and tunnelling.

Author Response

Thank you for your suggestion, please see the attachment.
